# Biological and Clinical Aspects of Metastatic Spinal Tumors

**DOI:** 10.3390/cancers14194599

**Published:** 2022-09-22

**Authors:** Jakub Litak, Wojciech Czyżewski, Michał Szymoniuk, Leon Sakwa, Barbara Pasierb, Joanna Litak, Zofia Hoffman, Piotr Kamieniak, Jacek Roliński

**Affiliations:** 1Department of Clinical Immunology, Medical University of Lublin, Chodźki 4A, 20-093 Lublin, Poland; 2Department of Neurosurgery and Pediatric Neurosurgery, Medical University of Lublin, Jaczewskiego 8, 20-090 Lublin, Poland; 3Department of Didactics and Medical Simulation, Medical University of Lublin, Chodźki 4, 20-093 Lublin, Poland; 4Student Scientific Association at the Department of Neurosurgery and Pediatric Neurosurgery, Medical University of Lublin, Jaczewskiego 8, 20-090 Lublin, Poland; 5Student Scientific Society, Kazimierz Pulaski University of Technologies and Humanities in Radom, Chrobrego 27, 26-600 Radom, Poland; 6Department of Dermatology, Radom Specialist Hospital, Lekarska 4, 26-600 Radom, Poland; 7St. John’s Cancer Center in Lublin, Jaczewskiego 7, 20-090 Lublin, Poland; 8Student Scientific Society, Medical University of Lublin, Al. Racławickie 1, 20-059 Lublin, Poland

**Keywords:** spine, metastasis, tumor, cancer

## Abstract

**Simple Summary:**

Spine metastases are a common life-threatening complication of advanced-stage malignancies and often result in poor prognosis. Symptomatic spine metastases develop in the course of about 10% of malignant neoplasms. Therefore, it is essential for contemporary medicine to understand metastatic processes in order to find appropriate, targeted therapeutic options. Our literature review aimed to describe the up-to-date knowledge about the molecular pathways and biomarkers engaged in the spine’s metastatic processes. Moreover, we described current data regarding bone-targeted treatment, the emerging targeted therapies, radiotherapy, and immunotherapy used for the treatment of spine metastases. We hope that knowledge comprehensively presented in our review will contribute to the development of novel drugs targeting specific biomarkers and pathways. The more we learn about the molecular aspects of cancer metastasis, the easier it will be to look for treatment methods that will allow us to precisely kill tumor cells.

**Abstract:**

Spine metastases are a common life-threatening complication of advanced-stage malignancies and often result in poor prognosis. Symptomatic spine metastases develop in the course of about 10% of malignant neoplasms. Therefore, it is essential for contemporary medicine to understand metastatic processes in order to find appropriate, targeted therapeutic options. Thanks to continuous research, there appears more and more detailed knowledge about cancer and metastasis, but these transformations are extremely complicated, e.g., due to the complexity of reactions, the variety of places where they occur, or the participation of both tumor cells and host cells in these transitions. The right target points in tumor metastasis mechanisms are still being researched; that will help us in the proper diagnosis as well as in finding the right treatment. In this literature review, we described the current knowledge about the molecular pathways and biomarkers engaged in metastatic processes involving the spine. We also presented a current bone-targeted treatment for spine metastases and the emerging therapies targeting the discussed molecular mechanisms.

## 1. Introduction

Spine metastases are a common life-threatening complication of advanced-stage malignancies and often result in poor prognosis. Symptomatic spine metastases develop in the course of about 10% of malignant neoplasms [1].

The spine is the most frequent localization among the bone metastatic lesions and the third most common site of metastases after lungs and liver [2]. The thoracic region is the site of metastatic spine tumors in 60–70% of the cases, the lumbosacral spine (20–25%) and the cervical spine (10–15%) are less common [3]. In the analysis of CT scans, it has been observed that metastatic lesions typically occur in the posterior part of the vertebral body at first and then penetrate the pedicles [4]. Bone metastases most frequently develop from primary solid tumors (such as breast (70%), prostate (85%), lung (40%), and kidney (40%)) [5] (Figure 1). Women suffering from breast cancer with overexpression of the estrogen receptor or the progesterone-one receptor or HER2 triple-positive women and men with castration-resistant prostate cancer are most vulnerable to bone metastases [6].

Bone metastases can be osteolytic, osteoblastic, or mixed (osteolytic and osteoblastic components). According to Constans et al., osteolytic lesions constitute more than 70% of spinal metastases, 8% are osteoblastic, and mixed metastases (osteolytic and osteoblastic) account for 21% [7]. Osteolytic lesions are characteristic of metastases from breast cancer, lung cancer, and renal cancer, whereas metastases of prostate cancer most often form osteoblastic lesions. Causes of mixed lesions include many tumor types, but most frequently are observed in breast cancer [8].

Metastatic cells can reach the spine through different ways of dissemination—hematogenous (intravenous and arterial), by contiguity, and lymphatic. The intravenous route is the most common way of propagation and is carried out through the paravertebral venous plexus of Batson [9,10]. This venous system communicates the spine with intercostal veins of the pulmonary, caval, or portal systems and provides a direct route of dissemination for breast and prostate cancer to the spine [11].

The most common clinical symptom caused by metastatic lesions in spine is being refractory to severe treatment pain. Other symptoms include hypercalcemia, pathological fractures, and spinal cord or nerve root compression, which are referred to as skeletal-related events (SREs). The current treatment of spine metastases is mostly palliative. It focuses on the improvement of health-related quality of life (HRQoL) through the control of pain, protection, or improvement of neurological functions and maintenance of spinal stability [1]. To prevent SREs and increase the quality of life of patients with spinal metastases as a result, antiresorptive agents such as aminobisphosphonates and anti-RANKL monoclonal antibody denosumab have been approved by the FDA. These drugs act through inhibition of the osteoclast function, which leads to the interruption of the vicious cycle of bone metastases and increase of bone mass [12]. However, improvement of the overall survival (OS) and progression-free survival (PFS) by these agents suggested by some studies is highly questionable. Recently investigated novel therapeutic agents targeting specific molecular mechanisms in the bone microenvironment represent a promising way of treatment. The bone microenvironment constitutes a pivotal role in the process of spine metastases. Multiple molecular interactions between metastatic tumor cells and bone tissue result in the alteration of many molecular pathways, which leads to the development of osteolytic or osteoblastic lesions. Improving the understanding of these mechanisms may help in the development of novel drugs targeting specific biomarkers and pathways. Currently, multiple substances are being investigated in preclinical and clinical studies with regard to inhibition of the bone metastatic process.

In this literature review, we described the current knowledge about the molecular pathways and biomarkers engaged in metastatic processes involving the spine. We also presented a current bone-targeted treatment for spine metastases and the emerging therapies targeting the discussed molecular mechanisms.

## 2. Molecular Basis of Spine Metastases

### 2.1. Escape from the Primary Site

Tumor is a type of malignant cell growth where abnormal cells multiply uncontrollably with an ability for close or distant tissue invasion. Cells within a tumor may differ in many ways, e.g., in their proliferative potential or the ability to undergo apoptosis or metastasize [13,14,15,16]. The route that cancer cells must surmount to reach other organs is burdensome, with escape from the primary site being the first obstacle encountered [17,18,19]. Another essential step in the process of metastasis is the entrance of neoplastic cells into blood or lymphatic vessels through which they reach even remotely located organs [20]. Increased activity of the master regulator of angiogenesis—oxygen-sensitive transcriptional activator HIF (hypoxia-inducible factor-1)—contributes to both provision of the path utilized for metastatic spread as well as tumor cells’ nutrition via modulation of proangiogenic factors that include, e.g., VEGF (vascular endothelial growth factor), Ang-1, Ang-2 (angiopoietins), or PlGF (placental growth factor) [21,22,23].

A hallmark of metastasis, epithelial–mesenchymal transition (EMT) is a biological process where epithelial cells forfeit their adhesive properties and apical–basal polarity and acquire migratory as well as invasive features in order to transform into mesenchymal stem cells [24,25,26]. EMT is involved in both pathological (e.g., cancer formation) and physiological (embryogenesis, organ development, wound healing) processes [27,28] conditioned by biologically different EMT subtypes [29,30,31,32]. Reversibility of this process (MET—mesenchymal–epithelial transition) enables cancer cells to return to their original form in another organ [33,34].

Both EMT and MET comprise series of biochemical reactions regulated by a plethora of transcriptional factors including Snail/Slug, Twist, Six1, Cripto, TGF-β, and Wnt/β-catenin that, when activated, reprogram gene expression [35,36,37] and, in consequence, alter tumor cell properties. While the cytoskeleton undergoes remodeling, modified protein expression into proteins distinctive for mesenchymal cells, e.g., N-cadherin, FSP-1, α-SMA, α5β1 integrin, αvβ6 integrin, vimentin, type I collagen, laminin 5, and fibrotin lead to the termination of connection with the basal membrane [38,39,40,41]. As a result, cancer cells acquire enhanced capability for relocation. Additionally, enzymes produced by tumor cells that break down the extracellular matrix, MMP (matrix metalloproteinase) and ADAM (A disintegrin and metalloproteinase), enable vascular wall penetration [42,43]. EMT also provides resistance to apoptosis due to diminished cell adhesion. The same mechanism facilitates migration and, in turn, evasion of various factors that lead to cell destruction.

### 2.2. Cancer Cells Dissemination

Cancers can spread to bones through various pathways, both venous and arterial, as well as through the lymphatic route or by direct contact [44]. Although the lymphatic system is mentioned as a potential route for spread, the main routes to enter the spinal column are comprised of venous and arterial vessels [45]. The Batson plexus, a network of veins devoid of valves that connect pelvic and thoracic with intraspinal veins contributes to spinal metastasis. Due to the absence of valves, any increase in vena cava pressure is followed by increased blood flow within the plexus, leading to cancer cells dissemination. In turn, neoplastic metastases reach the vertebral body directly through nutritional arteries [46,47]. Less frequently, neoplastic lesions metastasize through direct contact, e.g., prostate cancer that metastasizes to the lumbosacral spine [48]. Additionally, tumor cells have the ability to adhere to blood cells, including platelets, which serves as protection from the detrimental effects of hemodynamic forces during flow [49,50,51], as well as to bone marrow cells produced by tumor mimic precursors of immune cells, which aids to avoid innate immune response [52].

### 2.3. Bone Invasion

There are two main theories of metastasis. One of them is the so called “seed and soil” theory proposed by Paget over 100 years ago. It says that cancers anchor where they find convenient conditions, similarly to seeds in fertile soil, a process not dependent on anatomical relations [52,53]. Ewing, in turn, said that metastases are only the result of the structure of the circulatory system and are strictly related to anatomical conditions, such as the diameter of the vessels or connections between organs. In the 1980s, the work by Hart and Fiedler on melanoma metastases confirmed that theory [54,55].

However, looking at the development of tumors, it seems that both theories are relevant. Bone marrow, due to abundant vascularization, constitutes a part of the bone tissue of relatively high affinity for tumor metastases. The bones of the axial skeleton (skull, spine, sternum, ribs, hips, shoulders) contain a substantial amount of red bone marrow and therefore are a frequent target of tumor spread [56]. Low velocity of the blood flow enables easier adhesion to endothelial cells and, in consequence, quicker integration with endosteum. Bone marrow and bone cells also produce cytokines, hormones, enzymes, as well as growth factors that regulate the immune system and affect the colonization of bone tissue by cancer cells [57,58].

Tumor cells release a plethora of factors [59,60], e.g., VEGFR1+—bone marrow-derived progenitor cells that activate VLA-4 that via binding to fibronectin [61,62,63,64] enables entrance to the potential metastatic site and create an appropriate environment, a premetastatic niche that facilitates tumor cells implantation [65,66,67,68,69].

The abovementioned mechanisms are best understood in an example of metastatic breast cancer. It has been found that CXCR4, also known as fusin, is necessary for breast cancer cells migration towards tissues that present a high quantity of its specific ligand—cytokine SDF1 (CXCL12). Among the organs that express high levels of SDF1 are lungs, liver, bone marrow, and brain, which explains the high affinity of breast cancer cells to these tissues [70,71,72,73]. Tyrosine kinase Src is activated through the binding of CXCL12 to CXCR4, and downstream effector AKT improves the survival of cancer cells that occupy bone tissues [74].

Primary tumor cells also produce substances that modify the extracellular matrix at sites of metastasis, e.g., lysyl oxidates. Conversely, exosomes and miRNAs are able to influence bone remodeling. Metalloproteinases (MMPs) and bone sialoproteins (BSPs) destroy basal membranes at the site of metastasis, stimulate angiogenesis, and activate various elements involved in the destruction of bone tissue and the spread of tumor cells [70,75,76,77].

Once cancer cells reach the bone marrow, their growth depends on multiple factors, including in situ vascularization, available space, type of bone remodeling, or proliferating potential of neoplastic cells [78].

Bone tissue is made up of three main types of cells: osteoblasts, osteoclasts, and osteocytes. Osteoclasts are cells that have the ability to dissolve and resorb bone tissue, while osteoblasts are responsible for the growth and remodeling of bone tissue. Processes that lead to the formation of bone metastases may have different mechanisms: osteolytic and osteoblastic. Sometimes both of these mechanisms work simultaneously. In a healthy organism, the activity of osteoclasts and osteoblasts corresponds to the RANK-RANKL/OPG system [79,80,81].

RANKL (receptor activator for nuclear factor κB ligand) is produced by the osteoblastic line and activated T lymphocytes [80,82]. It is responsible for activating the process of creating mature osteoclasts. While RANK (receptor activator for nuclear factor κB) is located on osteoclasts and serves as the main regulator during the formation of osteoclasts, RANK combines with RANKL, ligand of the receptor activator of nuclear factor kappa B (NF-κB), which at the same time causes upregulation of nuclear factor of activated T cells 1 (NFATc1) [83,84]. NFATc1 is the major regulator of cytokine expression in the process of osteoclastogenesis [85,86]. As a result of these changes, mature osteoblasts are formed. Their main task is old bone reabsorption, which causes the release of nutrients and creates space for osteoblasts. Osteoprotegrin (OPG) binds to RANK and blocks the formation of the RANK–RANKL complex, thereby inhibiting the maturation process of osteoclasts [87,88,89].

### 2.4. Osteocyte Physiology and Pathology

Osteocytes have an impressive lifespan of up to 25 years, during which they undertake several important physiological functions. They differentiate from osteoblasts with four stages of formation, type I preosteocytes (osteoblastic osteocytes), type II preosteocytes (osteoid osteocytes), and type III preosteocytes (young and old osteocytes) [90]. During the process of bone formation, the osteoblastic cell body reduces in size, and its cytoplasm expands, springing processes from out of the cell’s wall. The Golgi apparatus during the type I and II cycles has to be well-developed to efficiently synthesize type I collagen essential for maintaining the bone matrix. Entering the type III preosteocyte phase, the Golgi apparatus is reduced in size, and the osteocyte matrix proceeds from the incompletely mineralized phase to the formation of old osteocytes with high mineral density [90]. Mature osteocytes express such markers as DMP1, Sost, as well as the cx43 protein, which is believed to be critical in the role of keeping the cell from entering apoptosis [90,91]. It is theorized that cx43 influences bone cell activity by regulating the osteoprotegerin and sclerostin levels [91].

### 2.5. Osteoblast Physiology and Pathology

The aforementioned osteoblasts are generated from pluripotent mesenchymal stem cells that take on the crucial role of bone matrix synthesis by firstly establishing the collagen, OCN, osteonectin, BSP II, and osteopontin proteins alongside decorin and biglycan to create osteoids, which would be further mineralized [90].

Osteoblasts tend to communicate with osteocytes using the RANK–RANKL pathway as a way to order growth factor release from the bone matrix [92].

The aforementioned processes summarize the bone homeostasis. It has been established that the regions of the bone with the largest amount of turnover (trabecular bone) tend to become sites for metastatic cell growth. One very prominent factor of this cancer cell homing is CXC motif chemokine 12, also known as CXCL12 or SDF-1, produced by bone marrow stromal cells and osteoblasts, which was proven to be crucial to metastases [93,94]. Cancer cells interact with osteoblasts and osteoclasts, as well as the cytokines released by the bone in an otherwise physiological process [95].

### 2.6. Osteoblastic Metastasis Pathogenesis

Bone tissue is considered to be the third place in the aspect of frequency of metastatic changes. Most of these metastatic changes are the result of oncological diseases, primarily breast and prostate cancer [8,96]. The statistics analyzed in previous years showed bone metastasis to be the effect of up to 70–75% of breast and prostate cancers [17].

A factor that one has to take into consideration is the bone’s extreme metabolic activity derived from the three main cell types: osteocytes, osteoblasts, and osteoclasts. Osteoblasts account for 4–6% of total cells, osteocytes—90–95%, osteoclasts—about 1–4% [90].

Osteoblastic metastasis is characterized by the deposition of new bone rather than lysis of the already existing structures. It is most notably present in prostate cancers, carcinoids, small-cell lung cancers, Hodgkin lymphomas, and medulloblastomas [17]. Tumor cells invading the bone tend to produce growth factors such as bone morphogenic proteins, epidermal growth factors, and platelet-derived growth factors. There have been consistent data proving that the physical contact of osteoblasts and prostate cancer cells promote tumor growth in vitro via protein ECM components, proteoglycans (PGs), and junction-related molecules [97]. Some of the more prominent factors of this process are BMPSs, TGF beta, and endothelin-1. BMP4 has been proven to stimulate osteoblast differentiation after being secreted from PCa-118b prostate cancer cells via the pSmad1–Notch–Hey1 and GSK3 β–β-catenin–Slug pathways [98]. The aforementioned TGF beta, specifically, TGF beta 2, is secreted from the prostate cancer metastasized cells [98] to foster the progression of tumor growth [99]. Another mechanism of growth is the secretion of endothelin-1 which downregulates DKK-1 and stimulates the secretion of the Wnt signaling pathways proven to be associated with lytic lesions and suppressing the growth of bone tissue in myelomas [100]. This occurs because DDK1 inhibits the production of osteoblasts by preventing the binding of low-density lipoprotein receptor-related proteins 5 and 6 (LRP5/6) in osteoblast precursors [101].

The role of PTHrP fragments in the process must not be underestimated. The parathyroid hormone-related protein increases calcium absorption and bone resorption [102], but it greatly increases the metastatic growth of cancer cells [103]. It has been theorized that NH2-terminal fragments of this protein stimulate bone formation via the ETA receptor because of the shared sequence homology to ET-1, which was proven to increase metastatic growth [104].

Other research has proven that PTHrP acts as a mediator in osteoblastogenesis, increasing early osteoblast differentiation and proliferation of bone marrow cells [105].

One other crucial aspect of bone metastasis that has to be taken into consideration is the so-called vicious cycle. Prostate cancer cells that have been freshly metastasized tend to produce PDGF, ET1, and BMPs that activate osteoblastic differentiation and bone matrix formation. As mentioned before, bone turnover marks the sites where tumors tend to grow, as the freshly synthesized structures are rich with growth factors such as IGF, FGF, and TGF-β that attract prostate cancer cells [95]. The physical contact of tumor cells and osteoblasts further promotes the secretion of growth factors—the vicious cycle continues to propel itself until it reaches the physical limits of the metastatic site.

### 2.7. Osteolytic Bone Metastasis

When discussing osteolytic metastases, we must look at the research on breast cancer. The majority of breast cancer metastases are lytic; breast cancer cells produce TNF-A, IL-8, IL-11 [106], IGF1, LIF (leukemia inhibitory factor), lysyl oxidase [107], RANK ligands, and PTHrP (Figure 2) [108].

The receptor activator of nuclear factor kappa B ligand (RANKL) and the macrophage colony-stimulating factor (M-CSF) are considered essential for proliferation and differentiation of osteoclast progenitor cells [109].

PTHrP is an osteoclast-activating factor; it stimulates osteoblasts to produce RANKL, which binds to RANK-stimulating osteoclasts to increase bone resorption [110]. IGF-1 and TGF beta are then released from bones and further fuel PTHrP production [111].

TGF-β is produced by active osteoclasts and increases the production of PTHrP. The research on samples of MDA-MB-231 breast cancer cells with neutralized TGF-β shows a significant decrease in osteolytic lesions [112].

TGF-β has also been reported to activate Notch ligand Jagged1 [113] proven to be an important tumor growth stimulant and COX-2 expression [114] which, in turn, stimulates the production of PGE2 in MDA-MB-231 breast cancer cells, a proven factor in bone resorption. It is worth noting that binding to EP4 receptor PGE2 increases the number of RANK ligands produced by osteoblasts [115]. It has been proven to be prominent in estrogen receptor-negative breast cancer cells and their metastatic tumors [107]. Another way of initializing osteoclastogenesis independent from RANKL is the aforementioned lysyl oxidase, a mediator of HIF-1 and the cause of metastasis in hypoxic cells [116].

Leukemia inhibitory factor (LIF), which is part of the IL-6 family of cytokines, is a known breast epithelial growth suppressor [117] that also promotes breast cancer metastasis utilizing the AKT–mTOR signaling pathway in the absence of the lncRNA-CTP-210809.1 gene [118], which is, unlike the aforementioned lysyl oxidase, independent from the estrogen receptor status [117].

Bone-derived insulin-like growth factors are a family of abundant mediators generated by the bone. The IGF-1 receptor was proven to be present in up to 86.7% of the cases where metastatic cancer cells were analyzed. The studies conducted on IGF-IR-disabled mice models have shown decreased mitosis and increased apoptosis of metastatic sites formed by breast cancer, multiple myeloma, neuroblastoma, and prostate cancer [94]. IGFs tend to promote bone metastases utilizing the IGF-IR, Akt, and NF-kappa B pathways [119].

Tumor necrosis factor alpha is a well-studied proinflammatory cytokine and a very strong bone resorption inducer. TNF-α activates the AP-1 and NF-kappa B transcription factors, leading to NFATc1-mediated expression of osteoclast-specific genes. TNF-α can also activate RANKL expression directly, alongside IL-6 and IL-1, to increase bone resorption or, working via the DDK-1 protein, downregulate bone formation [120]. Other interleukins 8 and 11 are also essential to the process of osteolysis by stimulating osteoclast formation independently from the RANKL pathway by utilizing JAK/STAT3 osteoclastogenesis (Figure 3) [106].

## 3. Role of the Immune System

Due to the complexity of the processes involved in the formation of bone metastases, the part of the immunology which investigates, inter alia, the mechanisms of metastasis formation has been termed osteoimmunology and it still remains the subject of multilateral scientific research [121,122].

Many cells of the immune system participate in the formation of metastases, including macrophages, T cells, NK cells, dendritic cells, myeloid-derived suppressor cells (MDSCs) [57,121,122,123].

Bone marrow is a space rich in cells of the immune system, for instance, T CD4+ or T CD8++ lymphocytes. In osteoclastogenesis, T CD4+ lymphocytes play an important role, especially Th17 cells which interact with osteoclasts mainly through osteoprotegerin (OPG)/RANKL/RANK, resulting in an increased activity of osteoclasts [57,122]. Likewise, the cells of the tumor located in the primary site produce many factors, such as IL-1, IL-6, IL-11, PDGF, MIP-1α, TNF, M-CFS, RANKL, and PTHrP, directly stimulating osteoclasts to form osteoclastic-type lesions [57]. Among the T CD4+ cell population, there are regulatory T cells (Treg) responsible for the maintenance of homeostasis of the immune system, but their increased amount reduces the immune response to cancer [57]. In turn, T CD8 + lymphocytes play the opposite role, with the cytotoxic substances TNF-α and IFN-α destroying cancer cells. Hence, it can be concluded that maintaining the balance between lymphocyte populations may play a key role in preventing cancer [57,122].

After escape from the bone marrow and peripheral blood, monocytes differentiate into macrophages after colonization in multifarious tissues. Macrophages are an important component in tumor progression. First, in the primary site, tumor-associated macrophages (TAMs) are responsible for the evolution of angiogenesis, migration of tumor cells, and escape from the primary focus. In turn, when they reach the metastasis site, they become metastasis-associated macrophages (MAMs) and play the metastasis-promoting role, being in charge of colonization and tumor development [121]. In addition, TAMs can be divided into two lymphocyte populations: M1-like, which represents the tumor-suppressing activity by producing cytokines such as IL-1, IL-6, IL-23, IFN-α, IL-12 which activate cytotoxic T lymphocytes and NK cells to completely remove tumor cells, or M2-like, which leads to tumor promotion. The ratio of these lymphocyte populations may be an important prognostic factor in assessing the course of disease [57,121,123].

NK cells are also involved in the control of bone homeostasis. Cytotoxic cells participating in the tumor cells-killing process, thus reducing the amount of NK cells, can result in cancer progression [57,122]. However, their role in metastasis development is vague, because there is research exposing that NK cells bringing about melanoma cells grow [124].

Dendritic cells (APCs) play a very important role in preventing the development of tumors, insomuch as through the possession of antigen presentation; they have an ability to induce activation and proliferation of Th and Tc lymphocytes, which may be responsible for inducing an antitumor response. Their properties are used in an attempt to create vaccines against cancer. However, cancer cells as well as activation of other cells of the immune system can modify the properties of APCs, causing tumor progression [57].

Myeloid-derived suppressor cells (MDSCs) are a heterogeneous group of immune cells from the myeloid lineage generated in bone marrow that suppress innate and adaptive immunity. Ordinarily, this amounts to the transformation of immature myeloid cells to mature myeloid cells, such as macrophages, dendritic cells, and granulocytes. However, in pathological conditions, such as when a tumor develops in the body, the conversion described above is inhibited (which results in an impaired immune response), as well as further development of the tumor. In addition, the ability of MDSCs to differentiate into osteoclasts has been proven, which can contribute to bone destruction [57].

## 4. Diagnosis

Regarding clinical practice, the relevant aspect is finding the place of metastasis origin. As in almost any branch of medicine, interview is the most essential tool for a clinical doctor. Patients with bone metastases most often report soreness in the suspected metastatic sites [125]. Additionally, it is worth paying attention to nonspecific symptoms such as weakness, excessive sleepiness, or weight loss often associated with the advancement of the neoplastic process, multiple metastases, and duration of the disease [126]. Abnormalities related to the developing neoplastic process can also be found in basic laboratory tests, such as blood morphology, after determining the level of ions such as calcium or specific tumor markers, for instance, PSA in prostate cancer. Going a step further thanks to the advancements in radiology, a wide variety of imaging tests is available [127].

### 4.1. X-ray Imaging

The elementary, relatively inexpensive, and most commonly available imaging technology is the X-ray. For the diagnosis of bone metastases, it is important to find the site and the extent of metastatic changes. Since X-ray does not allow for precise three-dimensional imaging or determining the genesis of the lesions, it is not a test that is ultimately used to diagnose metastases. However, an X-ray can be significant as a screening test, e.g., for primary care physicians.

### 4.2. Computed Tomography—CT

Another method helpful in the diagnosis of neoplastic bone metastases is computed tomography (CT), which is now extensively available and is a noninvasive examination. In comparison with the widespread X-ray, it is more sensitive and allows for better visualization of lesions. This diagnostic method allows us to visualize all types of metastases, both osteolytic (they have a clearer outline than on a regular X-ray image) and osteoblastic (less visible bone trabeculae, blurred border between the cortical bone and cancellous bone) [128]. CT is likely to obtain a spatial image, which allows for a more accurate determination of the extent of metastases. However, it is less sensitive in cases of early metastases to the cortical part of the bone and does not allow adequate visualization of infiltrative bone disease in which bone marrow is involved. CT can be performed to plan a surgical procedure; the imaging helps to select the appropriate access during vertebroplasty [129].

### 4.3. Magnetic Resonace Imaging (MRI)

Magnetic resonance imaging is a highly specialized test that uses magnetic water molecules contained in the body and, more specifically, hydrogen atoms. MRI has great importance in the diagnosis of neoplastic metastases, especially in the case of metastases to the spine, in the diagnosis of which it is considered the test of choice. It is worth emphasizing that this is a noninvasive test with high sensitivity and specificity [130].

In the case of whole-body MRI, which images the total physical structure in one examination, the effectiveness of detecting cancer lesions is higher in comparison to CT or even scintigraphy. However, the availability of such MRI is rarer and the examination is associated with higher expenses [131].

### 4.4. Scintigraphy

Scintigraphy (gamma scan) is a diagnostic method of nuclear medicine that provides means to identify diseases of the skeletal system. This examination relies on introducing radioisotopes attached to pharmaceuticals into the organism, recording the decomposition of these chemical substances, and then presenting them graphically. Among the imaging tests performed to detect metastases, it is the most sensitive. Scintigraphy also enables depicting the entire skeleton in one examination [132]. However, the disadvantage of this method of tissue imaging is its low specificity, which means that the anomalous image acquired does not have to be caused by neoplastic metastases but, for instance, by a healing fracture or an inflammation of a different type [133].

### 4.5. Biopsy

The biopsy is an invasive method consisting of extraction of the bone tissue for microscopic evaluation and obtaining histopathological confirmation of the etiology of the lesions. This test may be helpful, especially when the patient presents only bone transformations and no tumor foci in other organs are found. By performing a biopsy, we are likely to obtain information about the primary tumor. Performing this type of diagnostics is also important to arrange the treatment because the histopathological examination evaluates the tumor-specific biomarkers for neoplasms, which may be needed to guide therapeutic recommendations. In the case of bone involvement, we perform core needle biopsy. Although biopsy is an invasive method and is associated with the risk of complications, transpedicular biopsy of the spinal column is relatively harmless, as it avoids the most important structures such as nerves, vessels, the lungs, and the spinal cord [134].

A promising method of biopsy is one under the control of CT or MRI, which minimizes the risk of complications and allows for more accurate extraction of material for examination [135].

### 4.6. Biomarkers

As a result of osteoblastic and osteoclastic transformations in bones, many chemical substances are released, which are termed bone turnover markers (BTMs) [136]. To detect BTMs, samples of blood serum or urine have to be tested. Bone turnover markers include those related to bone resorption and those associated with bone formation [137] (Table 1).

It can be required for the assessment of diagnosis, progression of changes, or qualification for appropriate treatment.

### 4.7. Gene Expression Profiles

One of the crucial components of metastatic growth is the specific gene expression profile. Studies have shown that metastases are closely related to primary tumors, sharing some of the mutations that start the cancerous process, but with extra mutations nonexistent in primary tumors. Research has shown that this process of obtaining diverging mutations takes place around 87% of the molecular time within a primary tumor [138].

When analyzing breast cancer gene alterations, researchers have discovered that the JAK2 and STAT3 pathway components undergo several nonsense substitutions, splice site mutations, and frameshift indels, resulting in the inactivation of the pathway both in ER+ and triple-negative cancers; even though the primary tumor cells lacked the aforementioned changes, it still led to metastasis [138].

Breast cancer metastases were also examined for genomic copy number imbalances (CNIs). This research showed that the most promising prediction factor was the copy number loss at 8p22. Copy number gains at 1q41 and 1q41.12 and loses at 1p13.3, 8p22, and Xp11.3 increase the risk of metastatic changes appearing specifically in the bone [139].

Another mechanism of activating metastatic traits may come from changes in epigenetic alterations, such as the case of FOXA1 mutations in prostate cancer. Forkhead box A1 is an essential transcription factor, that can undergo class 2 activating mutations to increase DNA affinity, inactivate TLE3, and promote metastasis via the aforementioned WNT pathway [140,141].

A recent study has begun trying to correlate specific locations of metastatic changes in the spine with identifiable differentially expressed genes (DEGs) via analysis of the protein–protein interaction in the bone tissue involved in lung, breast, and prostate metastatic tumors. The most prominent DEGs were the upregulated JUN and downregulated PCNA. JUN regulates gene expression of the RANK–RANKL system which is indirectly responsible for bone resorption via osteoclast differentiation. The lack of PCNA is hypothesized to inhibit the maturation of immunological cells, as well as the E2-dependent DNA repair process, leading to an unchecked differentiation process of the surrounding tissue. The other hub genes presented in the aforementioned studies such as HRAS and RHOC also play a crucial role in activating osteoclast cells [141].

Further study is required to better understand the genetic basis of metastatic changes preferring a certain type of tissue.

## 5. Bone-Targeted Agents

Bone-targeted agents (BTAs) are the most popular drugs used for patients suffering from spine metastases. BTAs can decrease the incidence of skeletal-related events (SREs) including pathological fractures, pain, and hypercalcemia, which are common problems in these patients [142,143,144,145]. In light of extending the overall survival of oncological patients with bone metastases due to more and more effective anticancer treatment, the role of BTAs increases significantly [146]. BTAs include two main classes of agents—bisphosphonates and RANKL inhibitor denosumab.

### 5.1. Bisphosphonates

The first BTAs approved by the FDA for the treatment of bone metastases were bisphosphonates. Apart from the treatment of bone metastases, bisphosphonates are commonly applied in the therapy of osteoporosis and Paget’s disease [147]. Currently, they are also investigated as a potential treatment for otosclerosis, osteoarthritis of the knee, inflammatory rheumatic diseases, and many others diseases [148,149,150]

Clodronate and etidronate are non-nitrogen-containing bisphosphonates and represent the first generation of these drugs. The second generation is known as aminobisphosphonates and includes pamidronate and ibandronate [145]. Zoledronic acid (ZA) represents the third-generation agents, also known chemically as aminobisphosphonates, and exhibits the highest affinity for bone and antiresorptive strength [12]. ZA was regarded as a standard agent for SREs prevention in patients with bone metastases for almost a decade [151,152]. Bisphosphonates are characterized by the ability to permanently bind to hydroxyapatite molecules, which, on the one hand, provide their high affinity to the bone in vivo; however, on the other hand, they may for many years remain sequestrated in patients’ bones [153,154].

Bisphosphonates achieve the antitumor effect by three mechanisms of action—anti-resorptive effect, immunomodulation, and direct interaction with tumor cells [12]. The antiresorptive activity of the first-generation bisphosphonates results from the formation of cytotoxic metabolites through incorporation into nonhydrolyzable ATP analogs, which leads to osteoclast apoptosis [155]. Aminobisphosphonates owe their antiresorptive properties to the inhibition of farnesyl pyrophosphate synthase (FDPS) and through their ability to inhibit the dissolution of hydroxyapatite crystals [155,156]. FDPS suppression leads to the accumulation of intermediate compounds of the mevalonate (cholesterol biosynthesis) pathway in osteoclasts such as dimethylallyl diphosphate (DMAPP) and isopentenyl diphosphate (IPP) [153]. Overproduction of DMAPP and IPP results in the formation of ATP analogs ApppD and toxic ApppI, respectively. Accumulating ApppI inhibits ATP-dependent protein kinases (which probably impairs the function of the EGF receptor) and causes osteoclast apoptosis through mitochondrial ANT blocking [157]. Moreover, inhibition of FDPS leads to blocking prenylation of small GTPase proteins, such as Rho, Rac, and Cdc42, crucial signaling proteins in osteoclasts, which play the main role in cytoskeletal organization, intracellular vesicles trafficking, membrane ruffling, and apoptosis [158]. These mechanisms impair osteoclast function and survival, which leads to cell apoptosis and decreased bone resorption [159]. Regarding immunomodulatory activity, aminobisphosphonates might induce apoptosis of macrophages related to osteoclasts and activate γδT cells, a subset of T cells characterized by antitumor properties [12,160]. However, activation of γδT cells is responsible for the acute-phase reaction, which can occur in 20% of the patients treated with aminobisphosphonates [161]. The third mechanism of action results from direct interactions with tumor cells, which include negative influence on proliferation, migration, invasion, and induction of apoptosis [12,162].

### 5.2. Denosumab

Denosumab is a fully human monoclonal IgG2 antibody that binds and inhibits the receptor activator of nuclear factor kappa B ligand (RANKL) acting like osteoprotegerin (OPG). That interaction results in the blockage of the RANK/RANKL axis responsible for the maturation of osteoclast precursors [163]. It leads to inhibiting the activation of osteoclasts and a decrease in bone resorption. Moreover, the RANK/RANKL pathway generates regulatory T cells, increases the production of cytokines, and induces chemoresistance in vitro [164,165]. Additionally, overexpression of the RANK/RANKL axis is commonly found in many tumors, including prostate, breast, cervix, endometrium, thyroid, esophagus, and stomach tumors [165]. Therefore, the use of anti-RANKL agents such as denosumab may potentially sensitize resistant tumors to immunotherapy, suggesting its possible antitumor effect [164].

In contrast to bisphosphonates, denosumab does not accumulate in bone tissue and its effect can be reversible after termination of the treatment [166]. Similarly to bisphosphonates, denosumab can also be used in the treatment of osteoporosis, but in lower doses and at shorter time intervals [167].

### 5.3. Treatment Effectiveness of BTAs

Many studies and meta-analyses showed that BTAs are effective in decreasing the incidence of SREs, delaying the time incidence of the first SREs and enhancing the patients’ quality of life [168,169].

The majority of studies demonstrated a significant superiority of denosumab over ZA in delaying the time to the first onset of SREs and the development of multiple SREs [170,171,172,173]. Moreover, denosumab more effectively reduces strong analgesic use and better improves the HRQoL rate than bisphosphonates [170,173].

However, their influence on the overall survival or disease-free survival has not been observed [170,171,174]. Interestingly, some retrospective studies on non-small-cell lung cancer (NSCLC) have shown that denosumab may improve the OS [175,176] due to the direct blocking of RANKL in NSCLC tumors, in which RANK and RANKL expression was identified. However, in a randomized open-label phase III trial evaluating the addition of denosumab to the standard first-line treatment in advanced NSCLC (SPLENDOUR trial), improvement of the OS after adding denosumab to standard first-line platinum-based doublet chemotherapy was not observed [177]. Furthermore, in patients with early-stage breast cancer treated with neoadjuvant therapy, RANKL inhibition did not improve disease-related outcomes (D-CARE trial) [178]. On the other hand, in the ABCSG-18 trial, adding denosumab to adjuvant aromatase inhibitor treatment increased disease-free survival in nonmetastatic breast cancer patients [179]. Pantano et al. suggested that the conflicting outcomes of the D-CARE and ABCSG-18 trials result from high heterogeneity of breast tumor cells, which can affect treatment effectiveness [180]. They also for the first time demonstrated the expression of RANK on the circulating tumor cells of breast cancer, which may identify a subset of patients sensitive to denosumab treatment. However, the direct in vivo antitumor activity of denosumab still remains under discussion.

Regarding adverse effects, bisphosphonates and denosumab present similar overall rates [170]. However, in the case of certain complications, such as pyrexia, acute-phase reactions, and renal impairment, denosumab displays a significantly lower occurrence in comparison to ZA [151,170,173]. On the other hand, osteonecrosis of the jaw is more commonly observed in patients treated with denosumab [181,182]. Furthermore, due to the stronger antiresorptive properties of denosumab compared with bisphosphonates, hypocalcemia is a more frequent adverse effect during treatment with denosumab [173,183].

Despite the common use of BTAs in clinical practice, the optimal duration of treatment has not been established. According to the ASCO guidelines and the NCCN Clinical Practice Guidelines, treatment with BTAs ends up when the patient has a substantial decline in his general performance status or in case of severe toxicity [184,185]. The optimal dosing interval of BTAs also still remains undetermined. Initially, studies suggested that BTAs should be administered intravenously every 3–4 weeks [186,187]. However, less intensive treatment (every 12 weeks) was shown to be noninferior in comparison to the standard schedule [188,189,190]. The currently conducted REaCT-HOLD BMA randomized study evaluates the noninferiority of 24-week BTA schedule compared with a 12-week schedule (NCT04549207). Changing a 4-week schedule to a 12-week one, and even a 24-week treatment schedule may be beneficial for patients due to a decrease in BTA-related adverse effects, especially renal impairment. Moreover, it may be more cost-effective for healthcare systems [191]. Moreover, in a retrospective cohort study, Alzahrani et al. observed that the greatest risk for SREs was during the first year of BTA treatment compared with the second and third years [191]. Those findings emphasize the appropriateness of increasing dosing intervals and reducing the time of BTA treatment.

Among bisphosphonates, the FDA and the EMA approved pamidronate disodium (90 mg intravenously every 3–4 weeks) and zoledronic acid (4 mg intravenously every 3–4 weeks) for bone metastases from breast cancer and multiple myeloma in conjunction with standard neoplastic therapy. ZA has also been approved for use in patients with bone metastases from other solid tumors (in the case of prostate cancer, only in the castrate-resistant type) [192]. Additionally, the EMA approved oral bisphosphonate ibandronate for bone metastases in patients with breast cancer [193]. Denosumab, at a dose of 120 mg subcutaneously every 4 weeks, has been approved by the FDA and the EMA for the prevention of SREs in adults with bone metastases from solid tumors, excluding bone metastases from multiple myeloma (FDA) [193].

### 5.4. Complications of BTA Usage

With the increasingly common long-term use of BTAs, the prevalence of side effects is also increasing. The most frequent complications induced by BTAs include impaired wound healing, osteonecrosis of the jaw, hypocalcemia, and atypical femoral fractures [185,194,195,196]. Moreover, acute-phase reactions and renal impairment have been observed in patients with prolonged use of bisphosphonates, especially ZA [197]. Steller et al. investigated the levels of growth factors in patients after antiresorptive treatment and observed a significant decrease in the EGF and TGF-β1 concentrations in the patients treated with ZA and lower levels of TGF-β1 in the patients on anti-RANKL therapy. These observations may explain worse wound healing in these patients [194]. In the case of denosumab treatment, the use of platelet-rich fibrin may be beneficial to overcome this problem.

Prevalence of osteonecrosis of the jaw is only 1.03% among the patients treated with intravenous bisphosphonates and 3.64% in the case of high-dose denosumab treatment [198]. Therefore, it is not a frequent but severe complication. The pathophysiological mechanism underlying osteonecrosis of the jaw is not yet identified. The proposed theories include impaired bone remodeling, decreased angiogenesis, and the role of inflammatory or infectious factors [199]. Furthermore, the risk of osteonecrosis development is directly proportional to the duration of antiresorptive treatment and the dosage used [200]. The current management algorithm varies from providing good oral care and eliminating risk factors such as periodontal disease or tobacco smoking [200,201,202] to intravenous antibiotics and radical oral surgery in advanced cases [203,204]. However, prophylactic and treatment guidelines have not been established; thus, treatment of this complication may be challenging.

To prevent another complication, hypocalcemia, supplementation of vitamin D and calcium is crucial, especially in the case of denosumab, which is the strongest antiresorptive agent among BTAs [192,204].

## 6. Bone-Targeted Radioisotopes

Targeted radioisotope therapy is currently applied for patients with diffuse metastases both for palliative therapy and to improve survival. In contrast to bisphosphonates and denosumab, radiopharmaceuticals target the osteoblastic parts of osteosclerotic metastases. In the past, β-emitters such as samarium-153 (153Sa), strontium-89 (89Sr), and phosphorus-32 (32P) were used for pain treatment [205,206,207]. However, their use was associated with hematologic toxicity due to deep penetration into the bone tissue. Radium-223 (223Ra), an α-emitter, showed a lower toxicity than β-emitting radiopharmaceuticals due to a less penetrating character of α radiation (about 80 μm) [208].

Ra-223, as a calcium mimetic, can be deposited by activated osteoblasts near metastatic cells due to its ability of binding to hydroxyapatite in newly formed bone [209]. High-energy radiation delivered by Ra-223 to adjacent cancer cells leads to their destruction, sparing healthy tissues from irradiation at the same time. In the ALSYMPCA trial, six doses of Ra-223 (50 kBq per kg) administered intravenously every 4 weeks for patients with castrate-resistant prostate cancer and symptomatic bone metastases improved the median overall survival (14.9 vs. 11.3 months; HR, 0.70; 95% CI, 0.58–0.83; *p* < 0.001), reduced SREs, and prolonged the time to the first SRE compared with a placebo [210]. Ongoing clinical trials focus on combining Ra-223 with chemotherapy and immunotherapy (NCT03230734, NCT03996473, NCT04071223). However, the clinical trials with Ra-223 combined with chemotherapy and immunotherapy conducted to date, such as ERA-223, showed unsatisfactory results [211].

Based on the results of the ALSYMPCA trial, treatment of castrate-resistant prostate cancer and symptomatic bone metastases without visceral metastases with Ra-223 got approval from the FDA and the first-category recommendation by the NCCN [212], whereas 89Sr and 153Sa have been approved by the FDA for control of the pain from bone metastases [213].

Treatment with Ra-223 is regarded as well-tolerated and safe; however, thrombocytopenia and aplastic anemia have been observed in some cases [214,215].

## 7. Emerging Targeted Therapies

As bone-targeted agents such as bisphosphonates and denosumab only delay or prevent SREs in patients with bone metastases, exerting insignificant and disputable influence on the overall survival, it is necessary to search for new molecular agents. With the development of knowledge about bone metabolism and physiopathology of bone metastases during the last few decades, many potential agents have emerged in preclinical studies as potential inhibitors of specific biomarkers essential for metastatic progression.

Everolimus, a mammalian target of rapamycin kinase (mTOR) inhibitor, suppresses metastatic progression in the bone through inhibition of tumor-induced osteoclastogenesis. Moreover, everolimus in combination with standard therapies has been approved for the treatment of HER2-negative breast cancer and hormone receptor-positive advanced breast cancer [216,217].

TGF-β induces molecular pathways, which promote tumor growth in the advanced stage of cancer progression [218]. Suppression of this biomarker reduced bone metastases in multiple preclinical studies on breast and prostate cancer [219]. Moreover, the TGF-β2 signaling pathway is involved in the formation of metastatic niches [220]. Therefore, TGF-β should also be considered a target of novel potential therapies. Recent clinical studies evaluated TGF-β inhibitors such as M7824, fresolimumab, and galunisertib [221,222,223]. The results of these studies regarding effectiveness are promising. Additionally, each of them was well-tolerated.

The Src kinase has been identified as the key factor for the development of bone metastases through the PTHrP-mediated effect on osteoclasts. Its overexpression in breast and prostate tumors is related to shortened survival and a greater risk of metastasis [224]. Dasatinib is an oral tyrosine kinase inhibitor with multitargeted activity against the Src family kinases (SFKs), platelet-derived growth factor receptor (PDGFR), BCR-ABL, and mast/stem cell growth factor receptor (c-KIT) [224,225]. Dasatinib inhibits osteoclasts activation and cancer metastasis to the bone. Moreover, this agent may reduce bone pain. Other potential anti-src agents evaluated in clinical trials include saracatinib and bosutinib [226,227].

Endothelin A antagonists, such as atrasentan and zibotentan demonstrated a promising potential for bone metastases-targeted therapy in preclinical studies [228,229]. However, a placebo-controlled phase III trial showed the ineffectiveness of atrasentan in delaying the progression of bone metastatic lesions in castrate-resistant prostate cancer [230]. Interestingly, a preclinical study on zibotentan demonstrated that the effectiveness of that endothelin A inhibitor considerably depends on androgen ablation [231].

Glycoprotein Dikkopf-1 (DKK-1), an endogenous WNT pathway antagonist, enhances osteoclastic activity through inhibition of osteoblastic differentiation and increasing levels of RANKL [232]. Studies showed that DKK-1 was overexpressed in prostate and breast cancers and in multiple myeloma bone lesions [233]. In a preclinical study in a murine model of breast cancer, elevated DKK-1 promoted osteolytic metastases and increased the number of osteoclasts [234]. In a phase IB multicenter dose determination clinical study, a humanized monoclonal antibody targeting DKK1, BHQ880 in combination with ZA and anti-myeloma treatment increased bone strength and density [235]. However, the prevention of SREs was not achieved.

Anti-sclerostin inhibitors, such as romosozumab, have been recently approved for the therapy of severe postmenopausal osteoporosis [236]. However, its antimetastatic properties have not been investigated. In a preclinical study, BPS804, a monoclonal antibody targeting sclerostin, enhanced bone density and strength and decreased the number of metastatic bone lesions [237].

CXCR4, chemokine, and also one of the CXCL12 receptors, play an important role in the modulation of colonization of the bone by metastatic cells. The CXCR4 inhibitors investigated in studies include plerixafor and pentixafor [238]. Interestingly, it has been suggested that plerixafor can reverse homing of the tumor cells disseminated to the bone marrow and return them to the bloodstream [239]. The homing of cancer cells to the bone may also be potentially suppressed by E-selectin antagonists such as uproleselan (GMI-1271), according to a preclinical study [240].

The agents for the therapy of bone metastases currently investigated in preclinical and clinical studies are summarized in Table 2.

## 8. Radiotherapy

Radiotherapy (RT) is another vital tool in spinal metastasis treatment for both neoadjuvant therapies following surgery and focal control of metastatic diseases. Due to continuous development of radiation oncology, nearly 50% of patients diagnosed with cancer undergo RT eventually in the course of treatment [247,248]. It is also utilized as a palliative method implemented in order to alleviate pain symptoms, improve quality of life, and attenuate the possibility of disease-accompanying pathologic fractures. RT might also be adopted with the aim to provide spinal cord decompression [249].

### 8.1. Types of Radiotherapy Modalities

The most common modalities utilized in spinal metastasis treatment are ERBT (external beam radiotherapy) and SRBT (stereotactic body radiotherapy) [250,251]. ERBT precisely delivers high-energy x-rays to the targeted tumor tissue using a computer-mediated and radiologic image-adjusted linear accelerator [252]. SRBT distributes a high irradiation dose to the target extracranial tumor tissue in one or few highly ablative fractions with the use of image-guided technology [253]. Nonetheless, SBRT is considered more precise and allows sparing of spinal cord tissue [254]. Table 3 presents a classification of the available radiotherapeutic methods (Table 3).

### 8.2. Mechanism of Action

Regardless of the type of modality, the mechanism of action of ionizing irradiation is based on the influence of an electromagnetic wave or radiation of particles elicited on cancer cells [255]. As mentioned previously, cell’s energy absorption leads to reactive oxygen species (ROS) formation as well as destruction of various intracellular molecules, including DNA strands [256,257]. Although both cancer and normal cells are capable of reparation, a cancer cell’s ability to restore damaged DNA is far less effective [17]. Furthermore, the higher proliferation rate of cancer cells over the normal ones provides a higher significance to irradiation [247]. Ultimately, inefficient reparation processes lead to the arrest of cancer cell cycle, senescence, and necrosis [258,259].

The beneficial effect of ionizing radiation on the underlying cellular mechanisms of irradiated bone cells is not fully understood. Nevertheless, human and animal studies provide evidence that irradiation catalyzes depletion of osteoclast-activating factor (OAF) formation and damage of OCs (osteoclasts) within the tumor mass. Thus, inhibition of bone resorption and promotion of new bone tissue formation consequentially reduce the occurrence of pathologic fractures.

Cancer-induced bone pain (CIBP) is caused by the alteration of biological equilibrium within the bone microenvironment by secretion of chemical and pain mediators by tumor cells. Nociception-associated proinflammatory interleukins (IL-1β, IL-6, IL-11), tumor necrosis factor (TNF-α), chemokines, e.g., monocyte chemoattractant protein (MCP-1), macrophage inflammatory protein (MIP-1α), CXCR4, as well as matrix metalloproteinases and anti-inflammatory cytokine-transforming growth factor (TGF-β) contribute to tumor growth and metastasis [8,260,261]. Tumor cells then release cytokines that aim to further stimulate osteoclasts and, in consequence, bone resorption. Novel literature delivers evidence that radiation treatment not only acts on OCs and OBs, but also changes the bone microenvironment, including by reduction of inflammatory cells and chemical pain mediators which, in turn, results in cancer-induced pain relief [262,263,264] (Figure 4).

### 8.3. ESTRO-ACROP Guidelines

Skeletal-related events (SRE) that include pathologic fractures, pain, or spinal cord compression aggravate quality of life of affected patients and demand immediate medical intervention. In order to alleviate the aforementioned symptoms, The European Society for Radiation and Oncology/Advisory Committee on Radiation Oncology Practice (ESTRO-ACROP) formulated guidelines that aim to assist clinicians in accurately diagnosing and managing spinal metastases with special attention directed to external beam radiation treatment. These guidelines were printed in two issues following clinical dichotomization into complicated and uncomplicated metastases that require altered treatment. Complicated metastases (one third of the cases) comprise cases that involve spinal fractures, focal neurological deficits, or an associated soft tissue mass that cause spinal cord or cauda equina compression [265,266,267].

#### 8.3.1. Uncomplicated Spinal Metastases

The guidelines recommend bone scintigraphy followed by CT, PET/CT, or MRI for diagnosis of symptomatic bone metastases. MRI should also be performed in the case of neural or soft tissues infiltration or compression. However, in the event of a metastatic tumor of an unknown origin or where molecular profiling improves the available treatment options, core biopsy is particularly advocated.

According to the authors, conventional radiotherapy with a single fraction of 8 Gy ought to be used in case of pain symptoms that are refractory to pharmacological treatment whereas a single-fraction-hemibody or wide-field irradiation should be performed in disseminated pain triggered by polymetastatic disease. Sustained post-RT pain symptoms are advised to be recurrently irradiated with an unmodified dose. The guidelines also address the issue of respective techniques implied in palliative pain treatment; however, due to lack of high-level evidence regarding superiority of one over another to date, each approach should be considered on the individual basis. However, the authors recommend 3D conformal image-guided radiotherapy for uncomplicated bone metastases and clinical target volume (CTV)-based RT in the event of soft tissue mass involved [266].

#### 8.3.2. Complicated Spinal Metastases

The guidelines recommend an altered approach in complicated spinal metastases. On the assumption of spinal cord compression, whole spinal column MRI (or when contrast-enhanced CT is contraindicated) should be undertaken within 24 h post-diagnosis. When the tumor origin site is yet undiscovered, further diagnosis should include appropriate laboratory tests as well as histopathologic examination through definitive surgery or image-guided biopsy.

Following diagnosis, dexamethasone (10–16 mg IV) should be administered and followed by tapering doses of oral dexamethasone over the course of 10–14 days. No evidence supports administration of higher steroid doses. Furthermore, high steroid doses are equivalent to the increased risk of gastric ulcers. Therefore, proton pump inhibitors are recommended, especially when NSAIDS are used concomitantly.

Patients with relatively long life expectancy (over 3 months) with identified spinal instability (evaluated with the Spinal Instability Neoplastic Score (SINS) (Table 4)), paraplegia lasting less than two days, and with a single spinal metastatic site should undergo urgent surgery (decompression with/without stabilization) and subsequent neoadjuvant irradiation. However, the patients who fail to meet these criteria should be immediately referred to RT and receive a single fraction of 8–10 Gy. Although in some circumstances reradiation is necessary, it ought not to be repeated sooner than 6 months after, with the overall biological effective dose (BED) that does not exceed 100–135.5 Gy. In order to treat neuropathic pain, apart from adequate pharmacologic treatment and neurostimulation, a single dose of 8 Gy for conventional RT is advocated.

Pathologic fractures should be treated through both surgery as well as postoperative RT, obviating cases of poor prognosis and low-performance status, where RT is suggested to suffice. Currently, compression fractures are considered to be favorable candidates for balloon kyphoplasty or percutaneous vertebroplasty. Regardless of that, a single dose of 8 Gy or five or ten fractions of 20–30 Gy are advised to be used in the event of pathologic fracture prophylaxis or when recalcification is predetermined. Equivalent RT doses are also used for bone-overgrowing tumor masses [267].

## 9. Immunotherapy

Immunotherapy revolutionized the standards of cancer treatment and brought new hopes for improving the quality of life and overall survival in oncological patients [268]. The use of immune checkpoint inhibitors (ICIs) represents the most successful type of immunotherapy for metastasis treatment [269]. The most common agents include CTLA-4 inhibitor ipilimumab and PD-1 inhibitors pembrolizumab and nivolumab [270]. However, these drugs showed sufficient efficacy for a minority of patients, and a great number of them developed resistance against the CTLA-4 and PD-1 inhibitors [271]. Thus, the current studies focus on blocking other signaling molecules influencing metastatic processes such as TIM3, LAG3, VISTA, TIGIT, or NKG2A [272,273,274,275,276]. Inhibition of the abovementioned immune checkpoints expressed by T cells increased their immune activity against tumor cells. Therefore, stimulating receptors which enhance the antitumor effect of T cells such as OX40, ICOS, CD40L, or CD27 leads to a similar effect [277,278,279,280]. The currently studied immunotherapeutic approaches to metastasis also involves adoptive cell therapy (ACT), application of CAR-T cells, and inhibiting protumor immune cells in the metastatic tumor’s microenvironment [271,281,282]. However, the bone microenvironment exhibits specific immune characteristics that distinguish bone from other metastatic sites [283,284]. Compared with other organs, bone is characterized by a decreased number of effective cytotoxic cells, a greater number of suppressive immune cells, and the presence of immune cells interactions with osteoclasts and osteoblasts [283]. Therefore, this may affect therapeutic outcomes after use of popular immune checkpoint inhibitors for bone metastases, which has been shown by some studies [285]. Moreover, due to that phenomenon, the presence of bone metastases may impair the efficacy of immunotherapy used for primary tumor treatment, e.g., of NSCLC, where the bone microenvironment provides a “shelter” for disseminated tumor cells [283,284,286,287]. However, emerging studies investigating interactions between the bone microenvironment and metastatic cells reveal further essential molecules and raise hope for overcoming immunotherapy resistance. Indeed, combinatorial therapy composed of anti-CTLA4 agent ipilimumab and TGF-β inhibitor suppressed the expansion of bone metastatic lesions [285]. In the recent studies, a growing body of evidence is observed for the increased clinical efficacy of ICIs after concomitant use of bisphosphonates or denosumab [165,288,289]. Moreover, combination immunotherapy with radiotherapy has also proven to be beneficial [290]. The currently ongoing clinical trials evaluate efficacy of the ICIs’ combination with denosumab, Ra-223, radiotherapy or chemotherapy (NCT03669523, NCT03996473, NCT05502315, NCT05378334, NCT03795207). However, the above data regarding clinical efficacy of ICIs for bone metastasis remain scarce and are represented mainly by small prospective and retrospective studies. Thus, conducting prospective randomized controlled studies is necessary to confirm these results.

## 10. Conclusions

Tumors often metastasize to the bone, therefore, an important task for contemporary medicine is to understand the processes that lead to such a state and find different therapeutic options. Thanks to continuous research, there is more and more detailed knowledge about cancer and metastasis, but these transformations are extremely complicated, e.g., due to the complexity of reactions, the variety of places where they occur, or the participation of both tumor cells and host cells in these transitions. The right target points in tumor metastasis mechanisms that will help us make a proper diagnosis as well as find the right treatment are still being researched. Nowadays, there are therapies for cancer metastasis to spine available that give patients a chance of starting treatment. Nevertheless, they are often burdened with numerous side effects and are not always as effective as we would expect. Their application still gives little chance of recovery from metastatic disease to the vertebral column. The more we learn about the molecular aspects of cancer metastasis, for example, about the EMT, osteolytic and osteoblastic mechanisms of metastasis, the easier it will be to look for treatment methods that will allow us to precisely kill tumor cells with good effectiveness and without side effects for the entire body. It is also important to create therapies that will allow us to affect the neoplasm at various stages in as many grip points as possible and give a chance for treatment event to patients with very advanced neoplastic disease, which is undoubtedly tumor metastasis to the spinal column.

## Figures and Tables

**Figure 1 cancers-14-04599-f001:**
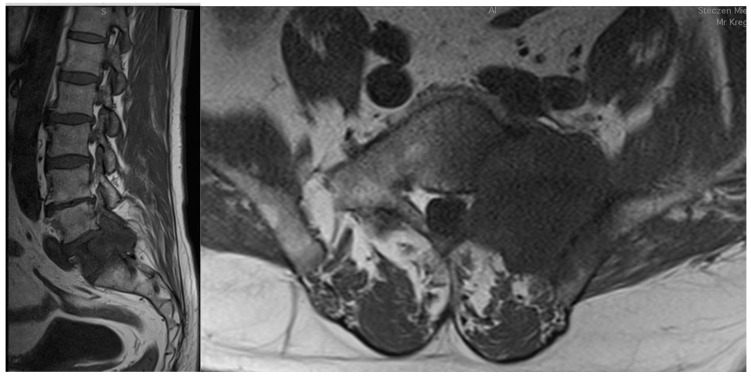
Sagittal and coronal T1-weighted MRI views of a patient diagnosed with sacral metastasis of a pulmonary squamous cell carcinoma.

**Figure 2 cancers-14-04599-f002:**
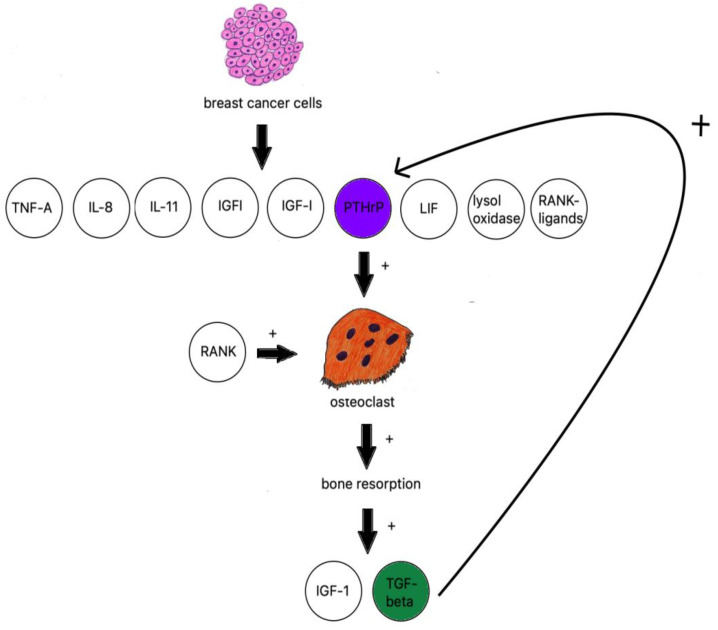
Breast cancer influence on osteoclastogenesis.

**Figure 3 cancers-14-04599-f003:**
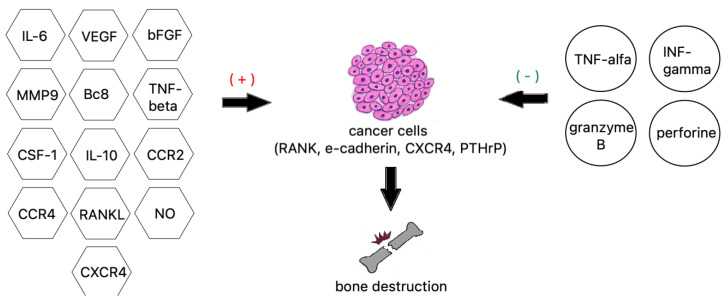
Factors contributing to bone destruction induced by metastatic cancer cells.

**Figure 4 cancers-14-04599-f004:**
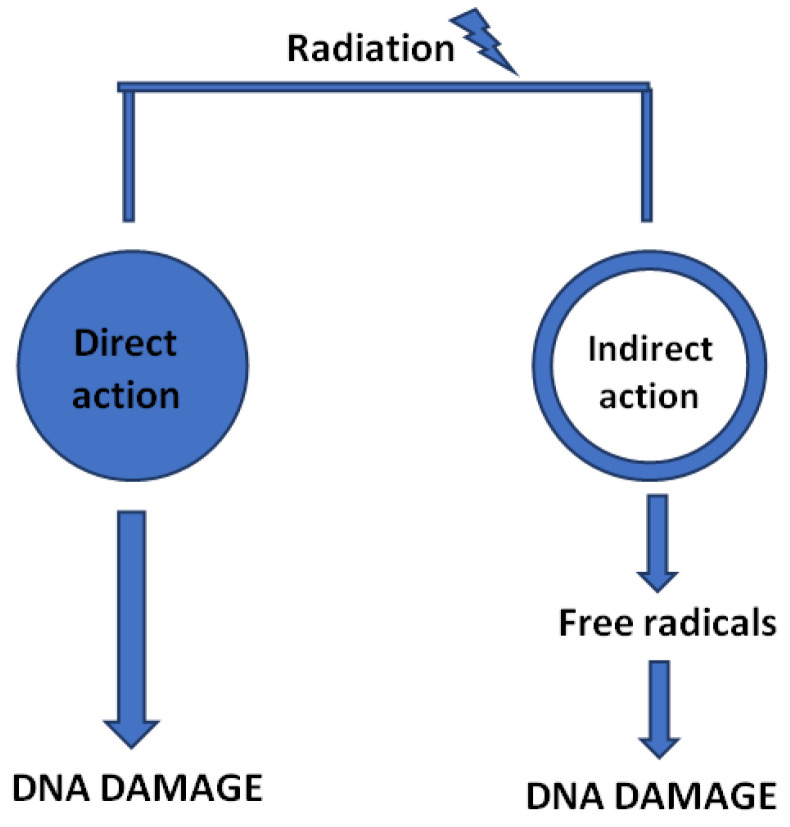
Direct and indirect influence of ionizing radiation leading to the damage of DNA strands.

**Table 1 cancers-14-04599-t001:** Bone resorption and bone formation markers.

Bone Resorption Markers	Bone Formation Markers
N-telopeptide of type I collagen (NTX), C-telopeptide of type I collagen (CTX), tartrate-resistant acid phosphatase (TRACP), receptor activator of nuclear factor-kB ligand/osteoprotegerin (RANKL/OPG), cross-linked carboxy-terminal telopeptide of type I collagen (ICTP), pyridinoline (PYD)	Procollagen type I N-terminal propeptide (P1NP), procollagen type I C-terminal propeptide (P1CP), bone alkaline phosphatase (BALP)

**Table 2 cancers-14-04599-t002:** The agents for the therapy of bone metastases currently investigated in preclinical and clinical studies.

Molecular Target	Drug	Antimetastatic Activity	Phase of Studies	References
mTOR	Everolimus	Reduction of lytic bone metastases;bone mass increase	Approved for clinical use	[216]
Endothelin A	Atrasentan	Analgesic effect	Phase 3 (NCT00134056)	[241]
	Zibotentan		Phase 3 (NCT00554229)	
Src kinase	Dasatinib	Inhibition of osteoclastic bone resorption; potential analgesic effect	Phase 2 (NCT00566618)	[227,242]
	Saracatinib		Phase 2 (NCT02085603)	
DKK-1	BHQ880	Bone mass increase	Phase 2 (NCT01302886)	[235]
E-selectin	Uproleselan	Blocking of metastasis extravasation and adhesion	Phase 2 (NCT04682405)	[240]
TGF-β	Fresolimumab	Disruption of the vicious cycle; reversion of the epithelial–mesenchymal transition; immune response enhancing	Phase 2 (NCT01401062)	[243]
	Galunisertib		Phase 1/2 (NCT02452008; NCT02672475)	
	M7824		Phase 1/2 (NCT04835896; NCT03524170; NCT03579472)	
Sclerostin	BPS804	Bone mass increase, decrease in the number of lytic bone metastases	Preclinical	[237,244]
CXCR4	Plexirafor	Reverse homing of tumor cells disseminated into the bone marrow	Preclinical	[238]
	Pentixafor			
Activin A	Sotatercept	Reduction of the CSC-like subpopulation; inhibition of the invasion, metastatic growth, and bone lesion formation	Preclinical	[245]
BMP pathway	DMH1	Reduction of the bone mass in osteosclerotic lesions	Preclinical	[246]

Abbreviations: BMP—bone morphogenetic protein; mTOR—mammalian target of rapamycin kinase; DKK-1—Dickkopf-1; TGF-β—transforming growth factor β; CXCR4—CXC chemokine receptor 4; CSC—cancer stem cells.

**Table 3 cancers-14-04599-t003:** Classification of the available RT modalities [3,25]

External Beam Radiotherapy (ERBT)	Internal Radiotherapy (Brachytherapy)
Conventional 2D external beam radiotherapy (cERBT)	Permanent implants
Three-dimensional conformal radiotherapy (3DCRT)	Temporary internal radiotherapy
Stereotactic body radiotherapy (SBRT)	
Charged particle radiotherapy (RT)	

**Table 4 cancers-14-04599-t004:** Spinal Instability Neoplastic Score (SINS).

Characteristic	Score
Location	
Junctional (O–C2, C7–Th2, Th11–L1, L5–S1)	3
Mobile spine (C3–C6, L2–L4)	2
Semirigid (Th3–Th10)	1
Rigid (S2–S5)	0
Pain	
Mechanical pain	3
Occasional pain, but not mechanical	1
Pain-free lesion	0
Bone lesion	
Lytic	2
Mixed	1
Blastic	0
Radiographic spinal alignment	
Subluxation/translation present	4
De novo deformity	2
Normal alignment	0
Vertebral body collapse	
>50% collapse	3
<50% collapse	2
No collapse with >50% vertebral body involved	1
None of the above	0
Posterior spinal element involvement	
Bilateral	3
Unilateral	1
None of the above	0

Total score and criteria: 0–6: stable; 7–12: potentially unstable; 13–18: unstable.

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
