# Peer review of "Biological and Clinical Aspects of Metastatic Spinal Tumors"

_cancers, 2022, doi:10.3390/cancers14194599_

Round 1

Reviewer 1 Report

This article presents a narrative review regarding selected aspects of bone metastases.

I kindly suggest that the paper in its present form should not be approved for publication, and I recommend major revision.

Explanation:

It's a really comprehensive review - congratulations! However, you stated in the title that you also aimed "clinical aspects".

Thus, I recommend you add sections regarding radiotherapy - it's probably the most common treatment used for bone mets:

- consider describing the mechanism of action of ionizing radiation applied to metastatic cells in bones,

- add some clinical data (to be more concise you may use existing guidelines, such as  https://doi.org/10.1016/j.radonc.2022.05.024  and 10.1016/j.radonc.2022.06.002

Author Response

Dear Reviewer,

Thank you sincerely for finding time to read and evaluate our article. In order to meet your expectations, we have added sections regarding radiotherapy and guidelines that you have provided. We hope that in this improved form, the article will suit high criteria to become published in Cancers.

All the best

Wojciech Czyzewski

Reviewer 2 Report

The manuscript describes molecular pathways, biomarkers engaged in metastatic processes to the spine, and bone-targeted treatment for spine metastases and emerging therapies targeting. The manuscript is a comprehensive collection of a variety of theories, parameters, methods, and procedures without a clear classification that it could contribute to a better assessment of possible biochemically based directions of targeted treatment. 

Main concerns:

The title is misleading the manuscript is not about spinal tumors but about metastases in the spine.

No section about the diagnostic methods for identification and characterization of the different localizations and molecular profiles of the metastasis.

The role of the immune system and local microenvironment is underrepresented. The emerging importance of the immunotherapies that can potentially target cancer metastasis should be also described.

Table x is not presented.

Literature numbers sometimes with space, sometimes without to the text.

Author Response

Dear Reviewer,

Thank you sincerely for finding time to read and evaluate our article. In order to meet your expectations, we have added sections regarding diagnosis, characterization of different localizations as well as molecular profiles of metastases. We have also elaborated on the role of the immune system and immunotherapies.

As the title of our article is “Biological and clinical aspects of metastatic spinal tumors” we decided not to change the title, unless you advice otherwise.

All the best

Wojciech Czyzewski

Round 2

Reviewer 1 Report

The manuscript has improved - thank you for incorporating my comments.

Please correct a typo in Figure 4 - "damage" rather than "demage".

Then, in my opinion, it will be suitable for publication.

Reviewer 2 Report

All concerns and questions were clearly answered.